Integrated genomic analysis of the stemness index signature of mRNA expression predicts lung adenocarcinoma prognosis and immune landscape

Lu Xingzhao 1 2
Du Wei 1
Zhou Jianping 1
Li Weiyang 1
Fu Zhimin 1
Ye Zhibin 1
Chen Guobiao 1
Huang Xian 1
Guo Yuliang 1
Liao Jingsheng liaojingsheng007@163.com 2
1 Thoracic Surgery Department, The Tenth Affiliated Hospital of Southern Medical University, Dongguan Institute of Clinical Cancer Research, Affiliated Dongguan People’s Hospital, Southern Medical University , Dongguan , China
2 Department of Medical Oncology, The Tenth Affiliated Hospital of Southern Medical University, Dongguan Institute of Clinical Cancer Research, Affiliated Dongguan People’s Hospital, Southern Medical University , Dongguan , China
Guan Fanglin
Electronic publication date: 2025 Feb 13
Publication date: 2025
Volume: 13
Electronic Location ID: e18945
Received 2024 Oct 29; Accepted 2025 Jan 16
Copyright: ©2025 Lu et al.
Copyright year: 2025
Copyright holder: Lu et al.
License: This is an open access article distributed under the terms of the Creative Commons Attribution License, which permits unrestricted use, distribution, reproduction and adaptation in any medium and for any purpose provided that it is properly attributed. For attribution, the original author(s), title, publication source (PeerJ) and either DOI or URL of the article must be cited.
License URL: https://creativecommons.org/licenses/by/4.0/

Keywords: Lung adenocarcinoma, mRNAsi, RiskScore model, Prognosis prediction, Drug sensitivity

Funding: The authors received no funding for this work.

==============================
mRNA expression-based stemness index (mRNAsi) has been used for prognostic assessment in various cancers, but its application in lung adenocarcinoma (LUAD) is limited, which is the focus of this study. Low mRNAsi in LUAD predicted a better prognosis. Eight genes (GNG7, EIF5A, ANLN, FKBP4, GAPDH, GNPNAT1, E2F7, CISH) associated with mRNAsi were screened to establish a risk model. The differentially expressed genes between the high and low risk groups were mainly enriched in the metabolism, cell cycle functions pathway. The low risk score group had higher immune cell scores. Patients with lower TIDE scores in the low risk group had better immunotherapy outcomes. In addition, risk score was effective in assessing drug sensitivity of LUAD. Reverse transcription-quantitative polymerase chain reaction (RT-qPCR) data showed that eight genes were differentially expressed in LUAD cell lines, and knockdown of EIF5A reduced the invasion and migration ability of LUAD cells. This study designed a risk model based on the eight mRNAsi-related genes for predicting LUAD prognosis. The model accurately predicted the prognosis and survival of LUAD patients, facilitating the assessment of the sensitivity of patients to immunotherapy and chemotherapy.

Introduction

Lung cancer is currently the most prevalent and deadly malignant tumors in the world (Thai et al., 2021; Hussain et al., 2024). Lung adenocarcinoma (LUAD) is the most common subtype of non-small cell lung cancer (NSCLC), accounting for about half of lung malignancies (Shukla et al., 2017; Cao et al., 2024). Heterogeneity and aggressiveness of LUAD results in a poor cancer prognosis, with a five-year survival rate lower than 15% (Li et al., 2019). At present, new immunotherapy methods for LUAD include molecular targeted therapy and immune checkpoint inhibitors, which have greatly improved the clinical treatment for LUAD patients (Song et al., 2022). However, the 5-year recurrence rate of LUAD patients with early intervention is still higher than 30% (Siegel, Miller & Jemal, 2020). Therefore, it is of great significance to accurately diagnose LUAD in an early and timely manner.

As one of the key components of tumor tissue, cancer stem cells (CSCs) can not only realize self-renewal and differentiation, but also have high heterogeneity and proliferation ability (Huang et al., 2020). CSCs play an important role in tumor progression, prognostic effect, immunotherapy benefit, and prognostic recurrence (Clara et al., 2020) in certain solid tumors (Chen et al., 2022). In a co-transplantation setting, CSCs have the ability to enhance the transformation of CD14+ peripheral monocytes into immunosuppressive M2 macrophages and facilitate the production of tumorigenic myeloid cells, which subsequently speeds up tumor growth in mice with compromised immune systems (Yamashina et al., 2014). Additionally, CSCs influence the recruitment and polarization of TH17 cells and regulatory T cells (Tregs) by releasing CCL1, CCL2, CCL5, and TGF-β, thereby contributing to an immunosuppressive milieu (Bayik & Lathia, 2021). At present, mRNAsi is an indicator that can accurately reflect the differentiation potential of tumor cells and help predict the possibility of cancer transformation in normal biological tissues (Weng et al., 2022). MRNAsi could be used to study the mechanism of tumorigenesis, cancer diagnosis and survival prediction in cancers such as endometrial cancer, acute leukemia, gastric cancer, glioblastoma, etc. (Liu et al., 2020b; Wang et al., 2021; Chen et al., 2020). As current research on the diagnosis and prognosis of mRNA expression-based stemness index (mRNAsi) in LUAD is insufficient, this study explored the clinical application potential of mRNAsi in LUAD.

Based on data acquired from The Cancer Genome Analysis (TCGA) and Gene Expression Omnibus (GEO) databases, we established a risk prediction model in combination with mRNAsi and clinical features. The model could be used to predict the survival outcome of LUAD, assess the sensitivity of patients to chemotherapy and immunotherapy, and better guide clinical treatment.

Materials and Methods

Data collection and preprocessing

(1) Data collection and preprocessing

The latest clinical follow-up information or expression data from 500 LUAD patient tissues with expression profiles of RNA-seq were sourced from TCGA database (Liu et al., 2020a). GSE30219 and GSE19188 data sets containing patients’ clinical survival data were obtained from GEO database (Barrett et al., 2013; Song et al., 2023). The mRNAsi of LUAD was acquired from a previous article (Malta et al., 2018).

(2) Data preprocessing

TCGA-LUAD samples without survival time and status and clinical follow-up information were deleted. Ensembl was transformed to the gene symbol, and the mean expression value of multiple Gene symbols corresponding to one gene was taken.

Samples from GSE30219 and GSE19188 were primarily processed as those in TCGA-LUAD. The probe was transformed into a gene symbol. Only the expression of multiple probes corresponding to a gene was averaged.

MRNAsi

Firstly, mRNAsi was acquired by using OCLR algorithm to analyze stemness characteristics of stem cells and their differentiated progenitors. MRNAsi for LUAD was acquired from a previous study to explore the association between clinical features and mRNAsi (Malta et al., 2018). High and low mRNAsi groups of LUAD patients were divided by the optimal mRNAsi value.

WGCNA

Limma package was used to draw volcano plot to analyze the differentially expressed genes (DEGs) between high and low mRNAsi under the threshold of |log 2 FC| > log 2 (1.5) and FDR <  0.05 (Sui et al., 2023). According to a total of 494 expression profiles of LUAD samples in TCGA database, the expression profiles of 1,929 DEGs were extracted, and the distance between each gene was computed using Pearson correlation coefficient. The Weighted gene co-expression network analysis (WGCNA) analysis was performed using R software to generate scale-free networks (Nangraj et al., 2020). Next, topological matrix was converted from expression-based adjacency matrix. Average-linkage hierarchical clustering method was performed with 30 gene per network as the minimum gene number when using mixed dynamic shear tree. Eigengenes of each module were calculated after determining gene modules by dynamic shearing method, followed by performing cluster analysis with deepSplit = 2, mergeCutHeight = 0.25, minModuleSize = 60. Close modules were combined into a new one. The correlation between mRNAsi and each module was further analyzed.

Construction of an mRNAsi-related risk model

The glmnet in R package was employed in univariate cox regression analysis and lasso regression analysis to reduce gene number in the model and extract key genes (Guo et al., 2021).

The following formula was utilized to compute the RiskScore for each patient: RiskScore= ∑βi×Expi

where i is the expression of a key gene and β is the Cox regression coefficient of the corresponding gene.

Low and high risk groups were divided by the median risk score. Kaplan–Meier (KM) method was used to draw survival curves. The timeROC in R package was employed to perform receiver operating characteristic (ROC) analysis on the risk score for prognostic prediction (Zhang et al., 2022b). Prognostic prediction for 1, 3 and 5 year(s) was assessed using the training dataset.

Screening and enrichment of DEGs between the two risk groups

The DEGs between the two groups were filtered using limma package under |log 2 FC| > 1 and FDR < 0.05. The R software WebGestaltR (V0.4.4) package was used for KEGG analysis and GO enrichment analysis between the two risk groups (Liao et al., 2019).

Immune microenvironment differences

According to immune cell genes in a previous study (Charoentong et al., 2017), ssGSEA algorithm was used to score the relative enrichment of 28 immune cell subtypes between the two risk groups. The overall immunity scores were calculated by ESTIMATE algorithm for of the TCGA data.

Immunotherapy and drug sensitivity analysis

TIDE software was used to access the potential clinical effect of immunotherapy on the two risk groups (Jiang et al., 2018), with a higher TIDE score suggesting greater immune escape and less immunotherapy benefit. The correlation between risk score and TIDE, Dysfunction and Exclusion was also compared.

The genomics of drug sensitivity in cancer (GDSC) stores the data on drug sensitivity (http://www.cancerrxgene.org). Spearman correlation analysis was employed to compute the correlation between drug sensitivity and risk sore with AUC in LUAD cell lines.

Nomogram

Risk subgroups were classified using a decision tree in terms of T. stage, N. stage, M. stage, Stage, risk score, age, and gender in the TCGA-LUAD cohort. Risk score and clinicopathological features were subjected to univariate and multifactorial cox regression analysis to determine significant prognostic factors. A nomogram was designed combining the risk score and other clinicopathological features for quantifying risk assessment and survival of LUAD patients. Calibration curve and Decision curve (DCA) were applied to assess the prediction performance and stability of the model.

Cell culture and drug treatment

In this study, human bronchial epithelial cells BEAS-2B (BNCC359274) as well as human LUAD cells NCI-H2009 (BNCC358037) and NCI-H1975 (BNCC340345) were purchased from BeiNa Culture Bio (Xinyang, China) and were used for the experiments. High-glucose DMEM (Cat. #31600, Solarbio, Beijing, China), Eagle’s Minimal Essential Medium (Cat. #LA10099, BALAB, Beijing, China), and RPMI-1640 media (Cat. #90023, Solarbio, China) were respectively used for culturing these cells. The culture media were additionally supplemented with 10% FBS (S9020, Solarbio, China). All cells were cultured in an environment with 5% CO2 at 37 °C.

Fresh medium containing 10 µM GSK2256098 (TargetMol, Cat. #T2281) was used to replace the existing medium for GSK2256098 treatment (Zhang et al., 2022a), during which the cells were incubated for 3 h.

Transient transfection

The EIF5A-specific small interfering RNA (si-EIF5A) and the corresponding negative control (si-NC) were obtained from GenePharma Co. located in Shanghai, China. These plasmids were introduced into H1975 and H2009 cells using Lipofectamine 3,000 reagents (Cat. #L3000-001) from Invitrogen (Carlsbad, CA, USA) in the USA. AGCAAGTTTTCTGAAACGTTTGG (si-EIF5A) was the target sequence for EIF5A siRNA. After a period of 48 h, the efficiency of the transfection was evaluated via qRT-PCR.

QRT-PCR

Total RNA was extracted from the cells (including LUAD cells and BEAS-2B cells) using TRIzol reagent (Cat. #15596-026, Thermo Fisher, Waltham, MA, USA). The quality of extracted RNA was determined by NanoDrop (Thermo Fisher, USA) to ensure the purity of RNA. To synthesize cDNA, we used HiScript II SuperMix (Cat. #R223-01, Vazyme, Nanjing, China) for the reverse transcription reaction. Each reaction system contained 500 ng of RNA, 2 µL of 5 × HiScript II SuperMix, RNAase-free water, and was adjusted to a final volume of 10 µL. The conditions of the reverse transcription reaction were as follows: pre-incubation at 25 °C for 10 min, reaction at 42 °C for 1 h, and termination of the reaction at 85 °C. The cDNA after reverse transcription was stored at −20 °C for backup. The qRT-PCR experiments were performed using SYBR Green fluorescent dye (Roche, Basel, Switzerland) for real-time quantitative PCR, and the experiments were performed in an ABI 7500 real-time quantitative PCR system (Thermo Fisher, Waltham, MA, USA). PCR amplification was performed under the following conditions: initial denaturation at 94 °C for 10 min; followed by 45 cycles: 94 °C for 10 s and 60 °C for 45 s. ACTB was acted as the internal reference. Experiments were performed in triplicate. Data results were normalized by the 2−ΔΔCt method and all data are expressed as mean ± standard deviation (SD). Statistical analysis was performed using GraphPad Prism 9.0 software. Table 1 displayed the sequence list of primer pairs.

Table 1 The primers of genes.

Gene	Forward primer sequence (5′–3′)	Reverse primer sequence (5′–3′)	
GNG7	CAAAGCGGCGTCTGACCTCATG	GGTTTCTTGTCCTTAAAGGGGTTC	
EIF5A	GAGCAGAAGTACGACTGTGGAG	CAGGTTCAGAGGATCACTGCTG	
ANLN	CAGACAGTTCCATCCAAGGGAG	CTTGACAACGCTCTCCAAAGCG	
FKBP4	TGACTCCAGTCTGGATCGCAAG	CTGGTTTGCAGGTGATGTGGCA	
GAPDH	CTGGGCTACACTGAGCACC	AAGTGGTCGTTGAGGGCAATG	
GNPNAT1	CCAACACATCCTGGAGAAGGCT	GGCTGACAACTCCAGTCTCTGT	
E2F7	TCTGAACCCGACTGTCCCTCTT	TTTGGCAGCCACATCCAGAGTG	
CISH	GCATAGCCAAGACCTTCTCCTAC	ACGTGCCTTCTGGCATCTTCTG	
ACTB	CACCATTGGCAATGAGCGGTTC	AGGTCTTTGCGGATGTCCACGT	

Transwell assay

Migration and invasion of H1975 and H2009 cell lines were detected by performing Transwell assay. Cells (5 × 104) were inoculated onto chambers coated (for invasion) or uncoated with Matrigel (BD Biosciences, Franklink Lakes, NJ, USA) (for migration). The upper layer and the lower layer were added with serum-free medium and complete DMEM medium, respectively. After incubation for 24 h, migrated or invaded cells were fixed using 4% paraformaldehyde (Cat. # P1110, Solarbio, China) and stained by 0.1% crystalline violet (Cat. #G1063, Solarbio, China). A light microscope (Thermo Fisher Scientific, Waltham, MA, USA) was employed for counting cell numbers.

Cell viability

CCK-8 kit (Cat. #CK-04, Dojindo Laboratories, Kumamoto, Japan) was conducted for measuring cell viability. Differentially treated cells were cultured at 1 × 103 cells/well in 96-well plates and added with CCK-8 solution at specific time points for 2-h incubation at 37 °C. Cell viability was indicated by the O.D 450 value of each well measured by a microplate reader.

Statistic analysis

All the statistical data were analyzed using R language (version 4.3.1) or GraphPad Prism 9.0. Specific analysis methods and software packages were mentioned in the above contents. Sangerbox (http://sangerbox.com/) offered auxiliary analysis in this article.

Results

The correlation between clinical features and mRNAsi in LUAD patients

We first looked at the correlation between mRNAsi and clinical characteristics of LUAD patients. There was no significant correlation between mRNAsi and stage (P = 0.064), N. stage (P = 0.29) and M. stage (P = 0.076) (Figs. 1B, 1D and 1E) but mRNAsi was significantly related to gender (P = 0.00027) and T. stage (P = 0.019) (Figs. 1A and 1C). In addition, compared to those with a high mRNAsi, patients with low mRNAsi tended to show a better prognosis (Fig. 1F), which provides important support for the application of mRNAsi-based prognostic prediction models in LUAD.

Figure 1 The correlation between LUAD stem cell characteristics and clinical characteristics.

(A–E) The relationship between mRNAsi and clinical features (T. Stage, N. Stage, M. Stage, Gender, Stage); (F) prognostic analysis of high and low mRNAsi patients in TCGA.

Constructing a weighted correlation network using mRNAsi-related DEGs

A total of 1,929 DEGs (1,402 down-regulated and 527 were up-regulated) were visualized in the volcano plot (Fig. 2A). β = 5 was selected to analyze the co-expression networks of different soft threshold capabilities (Figs. 2B and 2C). Gene dendrogram and module colors distinguished six modules, among which grey module was a gene set that cannot be classified into other modules (Fig. 2D).

Figure 2 Analysis of WGCNA based on mRNAsi-related DEGs.

(A) Volcano map of high and low mRNAsi differential genes; (B–C) Network topology analysis with different soft-thresholding powers; (D) Gene dendrogram and color of modules; (E) Correlation of each module with clinical information and aneuploid score; (F) Significant genes in the module.

Analysis on the correlation between each module and mRNAsi showed that the turquoise module was the one the most significantly correlated with mRNAsi (Fig. 2E). A total of 451 genes with high MM and GS were screened from the 775 module genes significantly correlated with mRNAsi in the turquoise module (Fig. 2F, MM > 0.4, GS > 0.4). These 451 genes were then used for subsequent studies.

Construction of an mRNAsi-related RiskScore model

A total of 345 genes showing greater prognostic impact were filtered from the 451 gene modules using univariate cox regression analysis. These genes were reduced by LASSO, and a mutual increase between lambda and the number of independent variable coefficients approaching 0 could be observed (Fig. 3A). A model was created by 10-fold cross-validation. The model was the optimal when lambda = 0.0458, under which a total of eight genes (GNG7, EIF5A, ANLN, FKBP4, GAPDH, GNPNAT1, E2F7 and CISH) were considered as the target genes in the following study (Figs. 3A–3B). The eight genes were subjected to multivariate Cox analysis and the risk coefficient for each gene was calculated (Fig. 3C).

Figure 3 Design and verification of mRNAsi related risk score model.

(A) The paths along which the coefficients of various independent variables change as the regularization parameter λ varies; (B) Analysis of the variation trajectory of λ during the regularization process; (C) Display of multivariable analysis results of risk score model genes; (D) ROC curve and KM survival curve of risk score in TCGA data cohort; (E) Risk score survival time and survival state and 8-gene expression in TCGA data cohort; (F) ROC curve and KM survival curve of risk score in GSE19188 data cohort; (G) ROC curve and KM survival curve of risk score in GSE30219 data cohort.

Finally, our risk model was designed: Risk score = −0.108*GNG7 + 0.154*EIF5A + 0.051*ANLN + 0.08*FKBP4 + 0.116*GAPDH + 0.229*GNPNAT1 + 0.06*E2F7 −0.042*CISH.

The KM survival curve showed a significantly lower survival probability of the high RiskScore group in TCGA than patients with a low risk score (P < 0.0001). The model had a high AUC value, as observed from 1-, 3- and 5-year prognostic ROC curves of the TCGA dataset (Fig. 3D). In TCGA dataset, the cutoff of risk score was used to divide high and low risk types. We found that survival was significantly better in high-risk patients than in low-risk patients. The expression of GNG7 and CISH showed a negative correlation with risk score, while that of six genes (EIF5A, ANLN, FKBP4, GAPDH, GNPNAT1, E2F7) was positively correlated with risk score (Fig. 3E).

Validation was further performed on LUAD datasets of GSE30219 and GSE19188 to assess the model’s robustness. Similar results in the two validation cohorts as in the training set were observed, with the ROC curves of both datasets having high AUC values. High risk score could predict a worse prognosis than a better prognosis of low risk score (Figs. 3F–3G).

Identified and enrichment of DEGs between high and low risk groups

In the TCGA dataset, a total of 400 DEGs (179 down-regulated and 221 up-regulated genes) between the two groups were found (Fig. 4A).

Figure 4 Identification and enrichment of DEGs in patients with different risk score groups.

(A) Volcano plot displaying the differentially expressed genes with high and low risk score groups; (B–C) GO functional enrichment analysis on the DEGs of high and low risk score groups.

The DEGs showing high and low risk score were subjected to KEGG and GO analyses. In GO analysis, we found that the top 10 biological processes and cellular components were all related to metabolism, cell cycle, and chromosome activity (Figs. 4B–4C).

Immune microenvironment differences between patients with high and low risk score

In the TCGA dataset, 13 types of immune cells in the low group with a better prognosis showed higher scores, while six types of immune cells in the high risk group with a worse prognosis had higher scores (Fig. 5A). Correlation analysis revealed a positive correlation between the risk score and infiltration of most immune cells (Fig. 5B). Innate and acquired immunity was better in the low risk group (Fig. 5C). We used ESTIMATE to assess the overall immunity score of the TCGA data, with the low risk group showing higher immunity scores (Fig. 5D). The MCP-counter immune score indicated that most of the immune cells with differences also scored higher in the low risk group (Fig. 5E).

Figure 5 Immune microenvironment differences among patients with different risk score.

(A) The distribution of 28 immune scores among high and low risk score groups in the TCGA; (B) Correlation between immune cell infiltration among different risk score groups; (C) Distribution of innate and acquired immunity among high and low risk score groups in TCGA; (D) Differences in the distribution of ESTIMATE scores among high and low risk score groups in the TCGA; (E) Distribution of MCP-counter immunity scores among high and low risk score groups in the TCGA.

Patients’ sensitivity to chemotherapy and immunotherapy assessed by risk score

The low risk group in the TCGA cohort showed lower TIDE scores, which indicated that those patients could benefit from taking immunotherapy (Fig. 6A). The dysfunction score was lower in the high risk group, while the exclusion score was lower in the low risk group (Figs. 6B–6C). At the same time, we computed the correlation between risk score and TIDE, dysfunction and exclusion, and the results indicated that our risk score was significantly positively correlated with TIDE and exclusion but significantly negatively correlated with dysfunction (Figs. 6D–6F).

Figure 6 Risk score assesses patients’ sensitivity to chemotherapy and immunotherapy.

(A–C) The distribution of TIDE, dysfunction and exclusion scores in the TCGA cohort was different among the high and low risk score groups. (D–F) TIDE, dysfunction, exclusion score and risk score correlation analysis; (G) Correlation between risk score and drug sensitivity in TCGA cohort in GDSC database; (H) AUC distribution of each drug based on the risk score model of the GDSC database.

Following the analysis on the correlation between drug sensitivity and risk score, GSK2256098C, Carmustine, Dacarbazine, PCI-34051 and Elephantin were selected as the sensitive drugs for LUAD patients (Fig. 6G). Patients in high risk group were more sensitive to Elephantin and GSK2256098C (Fig. 6H).

Figure 7 Design and validation of a clinical prediction nomogram based on mRNAsi-related RiskScore model.

(A) A survival decision tree was developed based on age, RiskScore, stage, gender; (B) Analysis of the overall survival differences between the three risk subgroups; (C–D) Comparative analysis between different groups; (E–F) RiskScore and clinicopathological features were subjected to univariate and multivariate cox analysis; (G) Nomogram model; (H) 1-, 3-, and 5-year calibration curve of the nomogram; (I) Decision curve of the nomogram.

Development and verification of a clinical prediction nomogram based on mRNAsi-related risk model

As shown in Fig. 7A, we constructed a survival prediction decision tree based on the risk model combined with clinical features (Stage) and evaluated the overall survival of LUAD patients by different risk groupings (C1, C2, and C3). Overall survival varied significantly in the three risk subgroups, with the highest survival probability in C1 and the lowest probability in C3 (Fig. 7B). Among them, all the patients in risk subgroup C2 were high risk patients, while all patients in group C1 were low risk patients (Fig. 7C). Patients’ survival status varied significantly in different risk subgroups, with the most favorable survival status in subgroup C1 and the worst survival status in subgroup C3 (Fig. 7D). The risk score was validated as the most significant prognosis factor (Figs. 7E–7F). Moreover, a nomogram integrating risk score and other clinicopathological characteristics was developed, and here risk score once again showed the strongest influence on evaluation LUAD survival (Fig. 7G).

Calibration curve reflected the prediction accuracy of the model. The predicted calibration curves for 1, 3, and 5 year(s) were close to the standard curve, which indicated that the nomogram could predict correctly (Fig. 7H). Additionally, significantly higher risk score and nomogram benefits than the extreme curve was observed, as shown in the DCA. Both the risk score and nomogram showed the strongest survival prediction (Fig. 7I). This further supports that our constructed risk model combined with clinicopathological features can effectively predict the survival prognosis of LUAD patients and provide potential clinical guidance value for individualized treatment.

Validation of the prognosis model using wet experiments

We experimentally validated the makers involved in the risk model using PCR experiments. The results of qRT-PCR indicated that the expression of GNG7, GNPNAT1 and CISH was downregulated in LUAD cell lines (H1975 and H2009). EIF5A, ANLN, FKBP4, E2F7 were upregulated in H1975 and H2009 cell lines, while GAPDH was not differentially expressed in BEAS-2B, H1975 and H2009 cells (Fig. 8A). We further explored the functions of the EIF5A due to its highest expression level in LUAD cells. The invasive and migratory abilities of LUAD cells were significantly decreased after suppressing EIF5A expression in H1975 and H2009 cells, indicating that EIF5A may promote LUAD progression (Figs. 8B–8C). Finally, we preliminary explored the effect of therapeutic drug GSK2256098. The mRNA expression of EIF5A was noticeably decreased in H1975 and H2009 cells after treatment with GSK2256098 at the concentration of 10 µM (Fig. 8D). Also, cell viability was significantly affected following the treatment of GSK2256098 at 10 µM (Fig. 8E). These results further indicate that the key genes identified based on mRNAsi characterization may have important biological functions in LUAD and also provide a rationale for their use as potential therapeutic targets.

Figure 8 EIF5A promotes the development of LUAD.

(A) RT-qPCR to validate the makers in the RiskScore model and perform relative quantitative analysis (n = 3). (B–C) Results of representative migration as well as invasion of H1975 and H2009 cells before and after inhibition of EIF5A and quantification of cell numbers (n = 3). (D) The mRNA expression of EIF5A in H1975 and H2009 cells before and after GSK2256098 treatment and relative quantitative analysis (n = 3). (E) Changes in cell viability of H1975 and H2009 cell lines before and after GSK2256098 treatment and quantification of O.D Value (n = 3). ns>0.05, *≤0.05, **≤0.01, ***≤0.001, ****≤0.0001.

Discussion

CSCs have a high degree of heterogeneity and strong proliferation and differentiation ability, and there are great differences in the heterogeneous characteristics of different cancers (Prasetyanti & Medema, 2017). CSCs play critical roles in cancer progression and invasion as well as in the development of drug resistance in tumors (Eun, Ham & Kim, 2017). The ability of CSCs to adapt to the environment is better than other tumor cells due to its efficient metabolism and self-regulatory biological processes that confer them drug tolerance including in immunotherapy (Najafi, Mortezaee & Majidpoor, 2019). Previous study showed that a circular RNA molecule in NSCLC could increase cancer cell stemness by affecting let-7 miRNA and PD-L1-related biological processes, promoting the ability of cancer cells to develop drug resistance and metastasis (Hong et al., 2020). MRNAsi is an index that can effectively reflect the stemness of cancer cells and has been therefore widely employed in the detection and prognostic evaluations of various cancers. The prognosis of head and neck squamous cell carcinoma could be correctly evaluated by marker prognostic feature genes based on mRNAsi in the research led by Tian et al. (2020). The present study explored the value of mRNAsi in the diagnosis and prognosis of patients suffering from LUAD.

Eight key genes (GNG7, EIF5A, ANLN, FKBP4, GAPDH, GNPNAT1, E2F7 and CISH) affecting prognosis were chosen based on mRNAsi in this study. The biological function and prognosis of these genes in a variety of cancers have been studied.

GNG7 as one of the key intracellular genes regulating division, growth, and apoptosis is low-expressed in different types of tumor tissues (Fang et al., 2022). Zhao et al. (2021) found that the up-regulation of GNG7 expression could suppress the activity of Hedgehog-related pathway and the expression of E2F1, preventing the deterioration and progression of LUAD (Zhao et al., 2021; Zheng et al., 2021). EIF5A is a key translation factor that regulates biological processes, for example, cell aging, metabolism, and apoptosis, and its overexpression in a variety of tumor tissues may be associated with tumor initiation and poor prognosis (Sfakianos, Raven & Willis, 2022; Coni et al., 2020). Martínez-Férriz et al. (2023) showed that EIF5A interacts with intracellular TFGB, increasing the protein translation efficiency of ribosomes and promoting the ability of LUAD to invade other normal tissues. ANLN is a key gene involved in mitosis and cell division process, and its high expression increases the growth capacity of tumor tissues (Cao et al., 2023). Previous study showed that inhibiting the expression of ANLN can noticeably promote the levels of proteins associated with intracellular apoptosis and immune and accelerate cancer cell death (Sheng et al., 2023).

FKBP51 is a molecular chaperone of glucocorticoids and its up-regulation promotes LUAD proliferation and deterioration through regulating the activity of IKK/NF-κB-related signaling pathways (Mangé et al., 2019; Zong et al., 2021). GAPDH is a multifunctional protein that plays different roles in different intracellular structures, among which nuclear GAPDH is closely related to cell division, apoptosis, and DNA structure and expression (Sirover, 2018). In a study of genes related to LUAD prognosis, Ouyang et al. (2023) found that GAPDH may be related to the ferroptosis mechanism of cancer cells, and that it may be the key to developing new targeted therapeutic drugs. GNPNAT1 is abnormally expressed in LUAD tissues, which can increase tumor cell stemness and promote the division and invasion of cancer cells (Zhang et al., 2021; Hu et al., 2023). CISH is a negative regulator of the action of the immune molecule IL-15 pathway in NK cells and is associated with immunosuppression (Zhu et al., 2020). We found that the eight genes were differentially expressed in LUAD cells, and that knocking down EIF5A decreased invasive and migratory abilities of LUAD cells. The mRNA expression of EIF5A was significantly downregulated in H1975 and H2009 cells and cell viability was reduced by therapeutic drug GSK2256098.

In this study, we screened key genes by mRNAsi and designed a model to help estimate the clinical survival status of LUAD patients. However, this study still had some limitations. Our data sample is only from public databases, and for this reason, in future studies, the sample size will be further expanded and data from LUAD patients from different regions and ethnicities will be combined for a more comprehensive validation. Furthermore, although the constructed model showed good predictive ability, further experimental validation was lacking to demonstrate the feasibility of its clinical application. In the future, we will select suitable LUAD xenograft mouse models for drug intervention and survival analysis to fully validate the practical application value of key genes in predicting patient prognosis.

Conclusion

In conclusion, the risk model proposed in this study demonstrates significant potential for prognostic prediction and treatment sensitivity assessment in patients with LUAD. In this study, combining mRNAsi and clinical features, eight key genes (GNG7, EIF5A, ANLN, FKBP4, GAPDH, GNPNAT1, E2F7, CISH) influencing LUAD prognosis were selected to develop a risk model. The risk model we constructed can effectively stratify the prognostic risk of patients and help identify patients with different risks, thus providing a clinical reference for individualized treatment. In particular, the model is able to assess the sensitivity of patients to different treatments (chemotherapy and immunotherapy), suggesting that low-risk patients may be more sensitive to immunotherapy. Our study may help physicians choose more precise and individualized treatment strategies based on patient-specific risk scores, thereby improving treatment outcomes and improving patient survival.

Supplemental Information

Supplemental Information 1 MIQE checklist

Abbreviations

mRNAsi mRNA expression based stemness index

LUAD Lung adenocarcinoma

TCGA The Cancer Genome Atlas

GEO Gene Expression Omnibus

ssGSEA single-sample gene set enrichment analysis

WGCNA weighted correlation network analysis

NSCLC non-small cell lung cancer

CSC cancer stem cells

EMT epithelial-mesenchymal transformation

DEGs differentially expressed genes

KM Kaplan–Meier

GDSC the genomics of drug sensitivity in cancer

DCA Decision curve

PD-L1 programmed cell death-Ligand

GNG7 G protein gamma subunit 7

EIF5A Eukaryotic initiation factor 5A

ANLN Anillin

FKBP51 FK506-binding protein 51

GAPDH glyceraldehyde-3-phosphate dehydrogenase

GNPNAT1 Glucosamine-phosphate N-acetyltransferase 1

CISH cytokine-inducible SH2-containing protein

DMEM-H Dulbecco’s Modified Eagle Medium-High glucose

EMEM Eagle’s Minimum Essential Medium

RPMI-1640 Roswell Park Memorial Institute Medium 1640

FBS Fetal Bovine Serum

TIDE Tumor Immune Dysfunction and Exclusion

OCLR Ordinary Constrained Linear Regression

MM Module Membership

GS Gene Significance

AUC Area Under the Curve

Additional Information and Declarations

Competing Interests

Author Contributions

Data Availability

The authors declare there are no competing interests.

Xingzhao Lu conceived and designed the experiments, performed the experiments, analyzed the data, authored or reviewed drafts of the article, and approved the final draft.

Wei Du conceived and designed the experiments, analyzed the data, authored or reviewed drafts of the article, and approved the final draft.

Jianping Zhou performed the experiments, prepared figures and/or tables, and approved the final draft.

Weiyang Li performed the experiments, authored or reviewed drafts of the article, and approved the final draft.

Zhimin Fu conceived and designed the experiments, analyzed the data, prepared figures and/or tables, and approved the final draft.

Zhibin Ye performed the experiments, authored or reviewed drafts of the article, and approved the final draft.

Guobiao Chen performed the experiments, analyzed the data, prepared figures and/or tables, and approved the final draft.

Xian Huang performed the experiments, prepared figures and/or tables, and approved the final draft.

Yuliang Guo conceived and designed the experiments, prepared figures and/or tables, authored or reviewed drafts of the article, and approved the final draft.

Jingsheng Liao conceived and designed the experiments, analyzed the data, authored or reviewed drafts of the article, and approved the final draft.

The following information was supplied regarding data availability:

The datasets generated and/or analyzed are available at GEO: GSE30219 and GSE19188.

The raw data is available in GitHub and Zenodo:

- https://github.com/jingshengliao/Raw-data.git.

- jingshengliao. (2024). jingshengliao/Raw-data: Raw data (v.1.1.0). Zenodo. https://doi.org/10.5281/zenodo.13989211.

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
