# Peer review of "Integrated genomic analysis of the stemness index signature of mRNA expression predicts lung adenocarcinoma prognosis and immune landscape"

_PeerJ, doi:10.7717/peerj.18945_

## Round 0.1 · original submission · Major Revisions

After careful consideration of your paper and the comments from three reviewers, we believe that major revisions are required before the manuscript can be considered for publication. Please address all reviewers' concerns comprehensively in your revision and provide a point-by-point response letter detailing the changes made.

Reviewer 1 ·

Basic reporting

These questions aim to probe deeper into the methodology, implications, and potential limitations of the study while encouraging authors to clarify their findings and enhance their research's impact in the field of oncology.
1. Can you provide more detailed explanations of the OCLR algorithm used to calculate mRNAsi? How does this method compare to other stemness scoring methods in terms of accuracy and reliability?
2. The study mentions data from 500 LUAD patients. How was this sample size determined, and do you believe it is sufficient to draw statistically significant conclusions about the prognostic value of the RiskScore model?
3. While you validated the RiskScore model using GSE30219 and GSE19188 datasets, can you elaborate on how representative these cohorts are of the broader LUAD population? Were there any demographic or clinical characteristics that could affect the generalizability of your findings?
4. The study lacks a comprehensive discussion on potential limitations. What specific limitations did you encounter during your research, particularly regarding data collection or analysis methods? How might these limitations impact the interpretation of your results?
5. You propose that the RiskScore can guide treatment decisions in clinical settings. What steps do you envision for implementing this model in routine clinical practice? Are there any barriers to its adoption that you foresee?
6. Can you provide insights into the biological roles of the eight identified genes (GNG7, EIF5A, ANLN, FKBP4, GAPDH, GNPNAT1, E2F7, CISH) in LUAD? How might these genes contribute to tumor biology and patient prognosis?
7. Your findings suggest that patients with lower TIDE scores benefit more from immunotherapy. Can you discuss how this insight could influence future clinical trials or therapeutic strategies for LUAD patients?
8. What future research do you propose based on your findings? Are there specific experiments or studies that could further elucidate the role of mRNAsi in LUAD or other cancer types?
9. Could you clarify the statistical methods used for analyzing correlations between mRNAsi and clinical features? Were any adjustments made for multiple comparisons to avoid false positives?
10. Were there any ethical considerations taken into account when using patient data from TCGA and GEO databases? How was patient consent handled in these datasets?

Experimental design

These questions aim to probe deeper into the methodology, implications, and potential limitations of the study while encouraging authors to clarify their findings and enhance their research's impact in the field of oncology.
1. Can you provide more detailed explanations of the OCLR algorithm used to calculate mRNAsi? How does this method compare to other stemness scoring methods in terms of accuracy and reliability?
2. The study mentions data from 500 LUAD patients. How was this sample size determined, and do you believe it is sufficient to draw statistically significant conclusions about the prognostic value of the RiskScore model?
3. While you validated the RiskScore model using GSE30219 and GSE19188 datasets, can you elaborate on how representative these cohorts are of the broader LUAD population? Were there any demographic or clinical characteristics that could affect the generalizability of your findings?
4. The study lacks a comprehensive discussion on potential limitations. What specific limitations did you encounter during your research, particularly regarding data collection or analysis methods? How might these limitations impact the interpretation of your results?
5. You propose that the RiskScore can guide treatment decisions in clinical settings. What steps do you envision for implementing this model in routine clinical practice? Are there any barriers to its adoption that you foresee?
6. Can you provide insights into the biological roles of the eight identified genes (GNG7, EIF5A, ANLN, FKBP4, GAPDH, GNPNAT1, E2F7, CISH) in LUAD? How might these genes contribute to tumor biology and patient prognosis?
7. Your findings suggest that patients with lower TIDE scores benefit more from immunotherapy. Can you discuss how this insight could influence future clinical trials or therapeutic strategies for LUAD patients?
8. What future research do you propose based on your findings? Are there specific experiments or studies that could further elucidate the role of mRNAsi in LUAD or other cancer types?
9. Could you clarify the statistical methods used for analyzing correlations between mRNAsi and clinical features? Were any adjustments made for multiple comparisons to avoid false positives?
10. Were there any ethical considerations taken into account when using patient data from TCGA and GEO databases? How was patient consent handled in these datasets?

Validity of the findings

These questions aim to probe deeper into the methodology, implications, and potential limitations of the study while encouraging authors to clarify their findings and enhance their research's impact in the field of oncology.
1. Can you provide more detailed explanations of the OCLR algorithm used to calculate mRNAsi? How does this method compare to other stemness scoring methods in terms of accuracy and reliability?
2. The study mentions data from 500 LUAD patients. How was this sample size determined, and do you believe it is sufficient to draw statistically significant conclusions about the prognostic value of the RiskScore model?
3. While you validated the RiskScore model using GSE30219 and GSE19188 datasets, can you elaborate on how representative these cohorts are of the broader LUAD population? Were there any demographic or clinical characteristics that could affect the generalizability of your findings?
4. The study lacks a comprehensive discussion on potential limitations. What specific limitations did you encounter during your research, particularly regarding data collection or analysis methods? How might these limitations impact the interpretation of your results?
5. You propose that the RiskScore can guide treatment decisions in clinical settings. What steps do you envision for implementing this model in routine clinical practice? Are there any barriers to its adoption that you foresee?
6. Can you provide insights into the biological roles of the eight identified genes (GNG7, EIF5A, ANLN, FKBP4, GAPDH, GNPNAT1, E2F7, CISH) in LUAD? How might these genes contribute to tumor biology and patient prognosis?
7. Your findings suggest that patients with lower TIDE scores benefit more from immunotherapy. Can you discuss how this insight could influence future clinical trials or therapeutic strategies for LUAD patients?
8. What future research do you propose based on your findings? Are there specific experiments or studies that could further elucidate the role of mRNAsi in LUAD or other cancer types?
9. Could you clarify the statistical methods used for analyzing correlations between mRNAsi and clinical features? Were any adjustments made for multiple comparisons to avoid false positives?
10. Were there any ethical considerations taken into account when using patient data from TCGA and GEO databases? How was patient consent handled in these datasets?

Additional comments

These questions aim to probe deeper into the methodology, implications, and potential limitations of the study while encouraging authors to clarify their findings and enhance their research's impact in the field of oncology.

Reviewer 2 ·

Basic reporting

The aim of this study was to mine prognostic biomarkers for assessing the drug sensitivity of lung adenocarcinoma (LUAD) based on the stemness index of mRNA expression. In this study, we firstly screened eight biomarkers in LUAD that were closely associated with its prognosis based on mRNAsi characteristics to establish a risk assessment model. The accuracy of the model in assessing the regulatory functions and immunoregulatory features of LUAD was also evaluated by enrichment analysis, metabolic analysis and immunoassay. The interaction between the model and cancer cell migration and invasion was subsequently verified by cellular experiments. The idea of this study is overall poor, but the following issues still need to be addressed before publication:
1. The drug sensitivity of this study was only briefly analyzed after constructing a prognostic model, so why is it the subject of this paper? Could a richer analysis be added to reveal key drugs targeting LUAD therapy? Please provide a rational explanation.
2. What is the correlation code between this feature of mRNAsi and immune regulation and metabolic regulation of cancer? Why did this study focus on analyzing these two components? Please give a description and enrich this aspect in the introduction section.
3. Does this paper have any relevance to the eventual prediction of drugs, given the relatively adequate analysis of the model in assessing immune cell infiltration and the effectiveness of immunotherapies? Or do these drugs actually modulate the immune microenvironment of LUAD? Please add to the results as well as the discussion.
4. In the cellular part of the experiment, is it too hasty to screen for target biomarkers based only on the significance of differential expression? Can further screening be done by bioinformatics analysis such as LASSO for example? Please rationalize and add analysis if necessary.
5. The Introduction section of this study did not systematically describe the current drugs acting on LUAD and the need for drug screening in this study, and it is recommended that a description be added.
6. Whether relevant studies based on mRNAsi mining of prognostic biomarkers for LUAD already exist and the shortcomings of these studies compared to the present study are suggested to be broadly described.
7. The introductory section explains that CSCs have an important role in cancer progression and prognosis, what are the specific roles they play? In particular, what are the regulatory effects on the activity of cancer cells? Specific descriptions are suggested.
8. The final module analysis in the WGCNA analysis was that the turquoise module was significantly positively correlated with the mRNAsi signature, but don't overlook the fact that the YELLOW module was the most significantly negatively correlated with the signature, so why wasn't this module followed up? Please provide explanatory notes.
9. For the eight prognostic biomarkers mined in this study, the discussion section is only a simple stacking based on existing reports without incorporating the results of this study, especially since this study has revealed the correlation of prognostic models with cancer immune profiles and drug-sensitive responses, please refine the discussion section by incorporating these results.
10. It is recommended that the description of limitations briefly state which cellular or tissue experiments are to be taken for subsequent in-depth exploration in this study to guarantee the subsequent sustainability of this study.

Experimental design

no comment

Validity of the findings

no comment

Reviewer 3 ·

Basic reporting

no comment

Experimental design

no comment

Validity of the findings

no comment

Additional comments

This study outlines an intriguing study that explores the utility of the mRNA stemness index (mRNAsi) in prognostic assessment for lung adenocarcinoma (LUAD). The work highlights the identification of eight genes linked to mRNAsi to construct a RiskScore model, offering insights into patient prognosis, immune response, and therapeutic sensitivity. Below are some constructive feedback points:
Line 141 - typo: “Gibco DEME F-12 medium” should be “Gibco DMEM F-12 medium”.
Line 185 – lacks citation: Figure 2E-F is listed in the figures but lacks citation in the manuscript.
In method section QRT-PCR, Much information is missing and cannot be copied, Analytical method? RNA how to converted cDNA?
The data concludes by emphasizing the RiskScore model's potential in prognosis prediction and therapy sensitivity assessment. To strengthen this, consider summarizing the key clinical implications explicitly, such as whether the model could guide personalized treatment strategies.
In the experimental section, the specific models of instruments and reagents used, along with their manufacturers' information and procurement sources, have not been detailed. To ensure the reproducibility of experiments and the reliability of results, it is recommended to supplement the following information: Clearly list the names of each experimental instrument, their manufacturers, models, and calibration status. For critical reagents, provide the commercial name, the full name of the supplier, and their product codes, particularly for antibodies, cell culture media, and primary chemical agents, among others. Furthermore, if specific software has been employed for data analysis, the software name, version number, and acquisition method should also be noted. Such enhancements not only increase the transparency of the research but also facilitate other researchers in precisely replicating experimental conditions and procedures.
The description of the results, in addition to a detailed list of the observed data and graphs, should indeed include a corresponding concluding statement in order to communicate the main findings directly and clearly to the reader.
There are many non-standard and consistent problems in the language expression of the article, which are embodied in the time expression, the naming and use of professional terms. For example, the time expression is sometimes used in the abbreviated form of "24h" and the hyphenated form of "2-H", which lacks a unified standard and should be unified into a standardized time expression form, such as "24 hours" or "2 hours", in order to maintain a consistent style and easy to understand. In addition, the concepts of "RiskScore" and "RiskType" mentioned in the article may be confused or not clearly defined. It is necessary to clarify the definitions of the two and ensure that these professional terms are used uniformly and accurately throughout the text to avoid misunderstandings caused by unclear expressions. In order to improve the quality and readability of the article, the whole article should be carefully checked to correct such problems of inconsistency and potential ambiguity.

---

## Round 0.2 · accepted · Accept

After thorough evaluation of your responses, I am pleased to inform you that your manuscript has been accepted for publication. Your responses comprehensively addressed the reviewer's concerns, particularly regarding Reviewer #1's comments about the OCLR algorithm methodology, the rationale for the 500-patient sample size, and the limitations of the GSE datasets. The detailed explanation of the eight identified genes' biological roles in LUAD and their underlying mechanisms provided strong scientific support for your findings. Additionally, your thoughtful discussion of the RiskScore model's clinical implementation pathway and the rigorous statistical methodology, including the appropriate use of Benjamini-Hochberg corrections, demonstrated the scientific merit of your work. These revisions have significantly strengthened the manuscript and enhanced its contribution to the field. Although Reviewer #1 was not available for a second review, your thorough responses to their initial concerns, combined with positive recommendations from the other two reviewers, provide sufficient basis for acceptance of your manuscript.

Reviewer 2 ·

Basic reporting

The aim of this study is to explore prognostic biomarkers based on mRNA expression and dryness index to evaluate drug sensitivity in lung adenocarcinoma (LUAD). In this study, we first screened eight biomarkers closely related to prognosis in LUAD based on mRNAsi features to establish a risk assessment model. The accuracy of this model in evaluating the regulatory function and immune regulatory characteristics of LUAD was also assessed through enrichment analysis, metabolic analysis, and immune assays. Subsequently, the interaction between the model and cancer cell migration and invasion was validated through cell experiments Although there have been many studies on the correlation between tumor stemness and tumor prognosis, this manuscript is generally innovative. However, the work is complete, the statistics are appropriate, and the results have been fully validated. Overall, it can be published.

Experimental design

no comment

Validity of the findings

no comment

Reviewer 3 ·

Basic reporting

no comment

Experimental design

no comment

Validity of the findings

no comment

Additional comments

This study outlines an interesting research that explores the application of mRNA dryness index (mRNAsi) in prognostic assessment of lung adenocarcinoma (LUAD). This work emphasizes the identification of eight genes related to mRNAsi to construct a RiskScore model, providing insights into patient prognosis, immune response, and treatment sensitivity. The author has done enough work, and the results and conclusions are solid.